# Harnessing Epigenetics for Breast Cancer Therapy: The Role of DNA Methylation, Histone Modifications, and MicroRNA

**DOI:** 10.3390/ijms24087235

**Published:** 2023-04-13

**Authors:** Joanna Szczepanek, Monika Skorupa, Joanna Jarkiewicz-Tretyn, Cezary Cybulski, Andrzej Tretyn

**Affiliations:** 1Centre for Modern Interdisciplinary Technologies, Nicolaus Copernicus University, 87-100 Torun, Poland; 2Faculty of Biological and Veterinary Sciences, Nicolaus Copernicus University, 87-100 Torun, Poland; 3Non-Public Health Care Centre, Cancer Genetics Laboratory, 87-100 Torun, Poland; 4International Hereditary Cancer Center, Department of Genetics and Pathology, Pomeranian Medical University, 70-204 Szczecin, Poland

**Keywords:** biomarkers, chemoresistance, epigenetic targets, histone deacetylase inhibitors (HDACi), DNA methyltransferase inhibitors (DNMTi), antagomiRs, mimic RNA

## Abstract

Breast cancer exhibits various epigenetic abnormalities that regulate gene expression and contribute to tumor characteristics. Epigenetic alterations play a significant role in cancer development and progression, and epigenetic-targeting drugs such as DNA methyltransferase inhibitors, histone-modifying enzymes, and mRNA regulators (such as miRNA mimics and antagomiRs) can reverse these alterations. Therefore, these epigenetic-targeting drugs are promising candidates for cancer treatment. However, there is currently no effective epi-drug monotherapy for breast cancer. Combining epigenetic drugs with conventional therapies has yielded positive outcomes and may be a promising strategy for breast cancer therapy. DNA methyltransferase inhibitors, such as azacitidine, and histone deacetylase inhibitors, such as vorinostat, have been used in combination with chemotherapy to treat breast cancer. miRNA regulators, such as miRNA mimics and antagomiRs, can alter the expression of specific genes involved in cancer development. miRNA mimics, such as miR-34, have been used to inhibit tumor growth, while antagomiRs, such as anti-miR-10b, have been used to inhibit metastasis. The development of epi-drugs that target specific epigenetic changes may lead to more effective monotherapy options in the future.

## 1. Introduction

Breast cancer is one of the most frequently diagnosed proliferative diseases in women worldwide, with an estimated number of 2.3 million new cases globally according to the GLOBOCAN 2020 data [1,2]. The risk of developing the disease is much higher in genetically predisposed women. There is no doubt that the genetic background is of key importance at every stage of the disease, starting with diagnosis and prognosis, monitoring the progression and choosing the right treatment protocol [3,4,5,6,7].

The vast majority (approx. 80–90%) of breast cancer cases are the so-called sporadic tumors. Up to 20% of diagnoses are hereditary cancers, associated with the presence of germline mutations, most often in the *BRCA1* (17q21.31) or *BRCA2* (13q13.1) genes. Approximately 5–15% of all diagnoses of the disease are cases with familial aggregation, for some of which the presence of mutations in the predisposition genes cannot be confirmed [1,8,9,10]. The most common cause of hereditary breast cancer are mutations in the *BRCA1* or *BRCA2* genes. Nevertheless, there are dozens of genes associated with the risk of breast cancer. Among them, genes participating in the repair of double-stranded DNA damage predominate. Mutations and polymorphisms in them can lead to abnormal cell growth, which can lead to the development of cancer. Among the genes that are associated with the development and progression of breast cancer, the following are currently listed [8,10,11,12,13,14,15,16,17,18,19,20]:*BRCA1* and *BRCA2* genes, which have the best documented association with breast cancer; having a mutation in these genes is responsible for a 50–80% risk of breast cancer and a 45% risk of ovarian cancer before the age of 85—with a mutation in the *BRCA1* gene and a 31–56% risk of breast cancer and 11–27% of ovarian cancer in *BRCA2* mutation [21,22,23,24,25,26,27,28,29,30];the *PALB2* gene, which is responsible for the repair of damaged DNA; carriers of the defective gene have a 35% risk of developing breast cancer before the age of 70 [31,32,33,34,35,36];the *CHEK2* gene, which is responsible for the production of a protein that inhibits tumor growth; women with a mutation in this gene have a twice as high a risk of developing breast cancer compared to the general population [37,38,39,40,41,42];the *NBN* gene, which encodes a protein regulating the DNA repair process and maintaining chromosome stability [43,44,45,46,47];*CDKN2A* gene, associated with the formation of proteins regulating the course of the cell cycle and inhibiting the growth of cancer cells [48,49,50,51,52];other, less known genes whose mutations may increase the risk of breast and other cancers, but which have not yet been precisely described, include *TP53*, *PTEN*, *CASP8*, *CTLA4*, *BARD1*, *BRIP1*, *CYP19A1*, *ATM*, *FGFR2*, *H19*, *LSP1*, *MAP3K1*, *MAP2K4*, *MRE11A*, *RAD51* and *TERT* [53].

According to current guidelines, breast cancer risk assessment should be based primarily on the analysis of 14 genes: *ATM*, *BRCA1*, *BRCA2*, *BRIP1*, *CDH1*, *CHEK2*, *NBN*, *NF1*, *PALB2*, *PTEN*, *RAD51C*, *RAD51D*, *STK11*, *TP53*. Genetic testing should be performed on every patient who: developed breast cancer before the age of 50, was diagnosed with triple-negative cancer, or developed ovarian cancer. In carriers of mutations in the *BRCA1* and *BRCA2* genes, the risk of developing breast cancer increases to even 80%. In addition, these *BRCA1*-associated breast tumors are usually triple-negative for estrogen receptor α (ER-), progesterone receptor (PR-) and HER2 (HER2-), which makes the development of targeted therapies difficult [21,22,23,24].

Detection of a specific mutation enables the implementation of appropriate further diagnostics and the development of individual and preventive recommendations, including imaging tests (ultrasound, magnetic resonance imaging, mammography). It has been shown that hereditary cancers often coexist in pairs: cancer appears in both breasts, it can develop in the ovaries, fallopian tubes, pancreas, and the risk of melanoma increases. Knowledge of the genetic background therefore enables effective and targeted monitoring of organs that are particularly vulnerable to the manifestation of the disease. Early detection of mutations enables the appropriate selection of diagnostic tests and providing the affected families with proper care and genetic counseling.

Genetic testing also plays an extremely important role in planning treatment. Conventional treatment protocols have greatly improved the management of BC patients, but subtype heterogeneity, the emergence of drug resistance, and disease relapse are major factors hampering the effectiveness of BC therapy. In the case of cancer patients, information on the genetic load changes the scope of surgery and adjuvant treatment. Knowledge about the genetic load is important for planning treatment, but also for the patient’s family, due to the high probability of the offspring inheriting the disease-causing mutation.

## 2. Epigenetic Regulations

In addition to the study of germline genetic changes, epigenetic changes affecting the modulation of predisposition gene expression, as well as causing disturbances in signaling pathways, especially those related to DNA repair, are being increasingly studied. Epigenetics is a critical mechanism for regulating transcription [54,55]. Epigenetic information is encoded in the structure and function of covalent modifications of DNA (hypomethylation or hypermethylation) and DNA-related nuclear chromatin histone proteins (such as acetylation, methylation, phosphorylation, sumoylation, ubiquitylation, or ADP-ribosylation). In the last few years, attention has been increasingly paid to the role of microRNAs as diverse expression regulators with high therapeutic potential. Epimutations can lead to the silencing of tumor suppressor genes independently and also in combination with pathogenic genetic mutations. Epimutations can also promote tumorigenesis by activating oncogenes. The events that lead to the onset of these epigenetic abnormalities are still not fully understood. Nevertheless, because epigenetic changes are inherited, they are selected in the rapidly growing population of cancer cells and provide a growth advantage for cancer cells, promoting their uncontrolled growth. For many genes of high predisposition, typical epigenetic changes have been determined, which affect the abnormal function of these genes and are one of the causes of cell transformation. Individual epimutations may act independently or together. As in the case of the *BRCA1* gene (Figure 1), they can be differentiated and involve different mechanisms. *BRCA1* alters epigenetics through physical interactions and transcriptional regulation of known epigenetic modifiers. In addition, as an E3 ubiquitin ligase, BRCA1 directly ubiquitylates histones.

Disturbances in the processes of epigenetic reprogramming of genomic DNA transcriptional activity may be the cause of neoplastic transformation. The fact that epigenetic aberrations, unlike genetic mutations, are potentially reversible and can be restored to their normal state by epigenetic therapy makes any epigenetic research promising and therapeutically relevant.

*BRCA1*/2-associated hereditary breast cancer is characterized by global changes in DNA methylation. In the presence of *BRCA1*/2 mutations, cancer cells are unable to properly repair DNA damage, leading to uncontrolled proliferation and tumor development. Changes in DNA methylation affect the expression of genes involved in DNA repair processes, cell signaling, proliferation, and differentiation, which further contributes to tumor development [57]. In the case of familial breast cancer not associated with *BRCA1*/2 mutations, changes in DNA methylation are also observed. It was shown that changes in DNA methylation were more correlated with familial breast cancer than with sporadic cases. These changes involved promoter regions of genes involved in cell cycle regulation, signaling, and apoptosis. In the case of sporadic breast cancer, changes in DNA methylation are more diverse and associated with the expression of different breast cancer subtypes. One study showed that DNA methylation patterns in breast cancer subtypes are so diverse that they can be used for classification [58].

It is believed that no two breast cancers are the same, and that the molecular profile of a breast tumor is different in each patient; therefore, it is very important to choose the right therapy. Unfortunately, in breast cancer, we still have few targeted therapies that directly target the changes that led to the cancer; however, the possibilities are constantly changing. The aim of this paper is to present the current state of knowledge on epigenetic factors and determining mechanisms in breast cancer, especially genetically determined, with particular emphasis on their impact on therapy. For many years, advanced research has been conducted on epigenome changes in breast cancer, and its goal is not only to understand the role of epigenetics in the development and progression of BC, but also to search for epi-drugs and develop new protocols based on epigenetic therapies.

## 3. Epigenetic Therapy in Breast Cancer

The reversible nature of epigenetic changes during oncogenesis has prompted the consideration of epigenetic therapy. Epigenetic therapy is a therapeutic approach that focuses on chemical modifications that affect gene activity in cancer cells. In breast cancer treatment, epigenetic therapy aims to restore the normal expression of tumor suppressor genes, which usually inhibit the growth of cancer cells. There are different types of epigenetic therapy, but the most commonly used in breast cancer treatment are histone deacetylase inhibitors (HDACi), DNA methyltransferase inhibitors (DNMTi) and RNA inhibitors [57,59]. Nevertheless, the search for effective epigenetic therapies has broadened and, in addition to these well-known inhibitors and target miRNAs, inhibition of HAT, class I, II, and IV specific HDACs, class III HDACs (sirtuins), KMT, KDM, and many kinase are being considered [60]. HDAC inhibitors inhibit the activity of enzymes that remove acetyl groups from histones, which leads to chromatin changes and activation of tumor suppressor genes. On the other hand, DNMT inhibitors inhibit the activity of enzymes that add methyl groups to DNA, which usually leads to the blocking of tumor suppressor gene expression. Inhibitors of epigenetic enzymes are referred to as “epi-drugs” [61,62,63,64,65]. RNA inhibitors, block the transcription and translation of mRNA, affecting the regulation of gene expression. This class of epigenetic regulators is most often used in the so-called replacement therapy [66].

Clinical studies have shown that epigenetic therapy can be effective in breast cancer treatment, especially in hormone-resistant breast cancer. Epigenetic drugs are being tested with increasing frequency, especially those effective in reversing DNA methylation and histone modification aberrations. In therapeutic concepts, it is important to take into account the interactions between various elements of the epigenetic machinery, and especially the synergistic effect of DNA and HDAC methylation inhibitors. In the case of microRNA expression disorders, therapeutic molecules (mimic or antagomiRs) are designed, whose task is to regulate the target miRs [66]. Combinatorial treatment strategies have been found to be more effective than individual therapeutic approaches. The first epigenetic drugs were tested for hematologic cancers.

## 4. DNA Methylation Aberrations

The first epimutations identified for the initial stages of tumor initiation and progression are profound changes in DNA methylation. A hallmark of the epigenome of cancer cells is genome-wide hypomethylation and promoter-specific hypermethylation of the CpG island promoter. The consequence of DNA hypomethylation is increased genomic instability, promoting chromosomal rearrangements. It also results in abnormal activation of genes (proto-oncogenes) and non-coding regions through various mechanisms that contribute to tumor development and progression.

Site-specific hypermethylation contributes to oncogenesis by silencing tumor suppressor genes. Such a silencing mechanism was confirmed for *BRCA1*. In addition to the direct inactivation of tumor suppressor genes, DNA hypermethylation can also indirectly lead to the silencing of various classes of genes by inactivating transcription factors and DNA repair genes [56,67]. Silencing DNA repair genes allows cells to accumulate further genetic changes leading to rapid tumor progression. The ability of DNA hypermethylation to silence predisposition genes in breast cancer is well-known, nevertheless, how the genes target this aberrant DNA methylation is not well-characterized. This mechanism is believed to provide a cell growth advantage, resulting in clonal selection and proliferation. Further understanding of how specific regions of the genome target DNA hypermethylation in cancer could potentially lead to the identification of additional therapeutic targets.

Hypermethylation most often concerns tumor suppressor genes however, it has also been confirmed for DNA repair genes, apoptosis, cell cycle regulation, cell growth, homeostasis and adhesion [62,68]. It is believed that DNA methylation status may be of value as both a diagnostic and predictive marker, including response to therapy. DNA methylation levels have been confirmed to be high at the gene loci proapoptotic genes (*HOXA5*, *TMS1*), cell cycle inhibitory genes (*p16*, *RASSF1A*) and DNA repair genes [69,70]. It has been shown that hypermethylation disorders of the genes encoding the estrogen receptor alpha and the progesterone receptor are correlated with the silencing of these genes and the development of ER- and PR-negative breast cancer. Hypermethylation of the *RASSF1A* gene is considered to be an important BC diagnostic marker, and the *PITX2* gene is considered a marker of tamoxifen resistance [62].

The relationship between *BRCA* and DNA methylation is the most widely studied [71,72]. The promoter of the *BRCA1* gene has more than 90 CpG islands, and the promoter of the *BRCA2* gene has 70 sites that can undergo methylation [73]. The *BRCA2* promoter is more often methylated; however, this gene is less frequently studied compared to *BRCA2* [73]. *BRCA1*-associated breast tumors show less DNA methylation compared to sporadic breast tumors [71,72,74]. *BRCA1* gene expression is significantly reduced or completely inhibited by the methylation of its promoter [75,76,77]. Hypermethylation of the promoter of this gene in sporadic breast cancer results in loss of function of this gene analogous to its mutation in hereditary cases [77,78]. CpG island promoter methylation is more common in young women with high-grade pathology and triple-negative breast cancer (negative estrogen receptor, negative progesterone receptor, negative ERBB2 [Her2/Neu]) [79,80,81,82]. In addition, mutated *BRCA1* is correlated with a number of epigenetic modifications in carriers. *BRCA1* physically interacts with the de novo methyltransferase DNMT3B, modulates heterochromatin [83], regulates the transcription of the methylation maintenance enzyme DNMT1 and prevents global DNA [67]. Genome hypermethylation in *BRCA1* carriers has been shown to be significantly correlated with cancer development [84]. Strong methylation of the estrogen receptor promoter alpha is the reason for the silencing the expression of this gene in familial cases of breast cancer [85]. Loss of function by mutated *BRCA1* leads to increased hypomethylation, which promotes proliferation and invasiveness, due to the increased expression of oncogenes such as *RAD9*, *c-Fos*, *H-Ras* and *c-Myc* [67,86].

Observations on the methylation status of genes involved in the initiation and progression of breast cancer have been used in attempts to improve the effectiveness of anti-cancer therapies. In vitro studies, animal models and clinical studies (Table 1) have shown the effectiveness of nucleoside analogs, e.g., cytidine analogs, i.e., azacytidine (5-aza-CR; Vidaza^®^, Celgene Corp., Summit, NJ, USA) and decitabine (5-aza-2′-deoxycytidine, 5-aza-CdR; Dacogen^®^, SuperGen, Inc., Dublin, CA, USA), in inhibiting DNA methylation in cancer cells [87]. Under culture conditions, after introducing the analogs, gene expression was induced, causing the differentiation of these cells. Cytidine chemical analogues are incorporated into the DNA of rapidly growing tumor cells during replication, and inhibit DNA methylation by forming covalent bonds with DNMT, leading to their depletion inside the cell [87]. The downregulation of DNA methylation induced by these drugs is responsible for inhibiting the growth of cancer cells by activating tumor suppressor genes that are abnormally silenced in cancer cells [88,89]. Studies have shown a reduction in tumor size in xenograft mice [90,91,92]. Tsai et al. [92] showed a correlation between the treatment of mice with azacytidine and the inhibition of tumor growth at the 5th week of therapy. Tao et al. [93] analyzed the possibility of breast cancer therapy using azacitidine. In their experiment, using the MCF7 and MDA-MB-231 cell lines, it was possible to inhibit 23 out of 26 hypermethylated genes in breast cancer. Moreover, for the *CLDN6*, *PRA*, *RIN1* and *VGF* genes, the ability to reactivate their expression was confirmed [93]. Efforts to treat breast cancer using decitabine, which prevents DNA remethylation, are also promising. Cai et al. [94] reported that this analog is able to activate the apoptosis-inducing ligand associated with tumor necrosis factor (TRAIL) in triple-negative breast cancer cells, thereby sensitizing cancer cells to chemotherapy. In addition, reductions in tumor growth have been observed in animal models [90,92]. *CTFR* is highly hypermethylated in breast cancer. This change was characteristic of invasive cancers. Low levels of *CFTR* protein correlated with poor patient survival. Treatment with decitabine enabled high expression of *CTFR* and thus inhibition of cell growth [95]. Borges et al. [90] noted the importance of aberrant methylation of the *PRKD1* gene promoter during tumor progression and lung metastasis. Restoring the normal methylation pattern of this gene with decitabine affected tumor aggressiveness and inhibited tumor spread in a *PKD1*-dependent manner. These researchers emphasized that the promoter status of the *PKD1* gene may be both a marker of early diagnosis and of significant importance in therapy.

Restoring the correct epigenetic state is also important for genes for which mutations have not been found, but they are inactivated in cancer cells. Genes such as *RASSF1*, *GSTP1*, *MGMT*, and *BRMS1* are markers of response to therapy, including difficult cases. Demethylation has also been shown to have promising effects in overcoming resistance to hormone therapy [62]. Restoration of expression of genes such as *ESR1* [96] or *PITX2* [97] sensitizes cancer cells to tamoxifen, and the analysis of promoter methylation status of these genes enables the identification of patients with positive expression of the hormone receptor that will not benefit from the drug administration [62].

**Table 1 ijms-24-07235-t001:** Exemplary clinical studies on the role of methyltransferase inhibitors in breast cancer (based on ClinicalTrials.gov (accessed on 5 March 2023) [98]).

NCT Number	Study Type	Description	Outcome Measures	Study Population
**Azacitidine**				
NCT04891068	Interventional	Determination of the effect of low-dose azacitidine therapy on tumor-infiltrating lymphocytes (TILs) in primary tumors from patients with high-risk early stage breast cancer.	Clinical response (change Ki67 and tumor size) of primary tumor following treatment with low-dose azacitidine therapy, DFS and OS measures.	Age ≥ 18 years of age at time of consent
NCT01349959	Interventional	Evaluation of the response rate using RECIST criteria of the combination of azacitidine and entinostat in women with advanced breast cancer, triple-negative and hormone-refractory.	Clinical Benefit Rate, OS, PFS, change in expression of relevant genes (e.g., *ER alpha* and *RAR beta*).	Histologically or cytologically confirmed adenocarcinoma of the breast triple-negative (ER-, PR-, HER2- or hormone-positive/ HER2-, with evidence of locally advanced and inoperable or metastatic disease).
NCT01292083	Interventional	Evaluation of the ability of DNA methylation inhibition using 5-azacitidine to induce expression of the ER and PR genes in solid human triple-negative invasive breast cancer.	Percentage of participants with ER/PR response after receiving 10 doses of 5-Azacitidine.	Resectable tumor measuring 2 cm or more; triple-negative invasive breast cancer
NCT02374099	Interventional	Assessing the efficacy and safety of the combination of fulvestrant with CC-486 in subjects with ER+, HER2- metastatic breast cancer who have progressed after prior AI.	Percentage of participants who achieved a confirmed CR, PR or SD to treatment, estimation of DoR and TEAEs.	≥18 years of age with metastatic breast cancer
NCT00748553	Interventional	Testing whether treatment of patients with advanced or metastatic solid tumors or breast cancer with Abraxane plus Vidaza is safe and results in good tumor response.	Percentage of participants with ORR measured using RECIST 1.0 criteria, including CR, PR, SD, or PD.	Patients with advanced or metastatic HER2-negative breast cancer who have not received treatment for their metastatic disease.
**Decitabine**				
NCT03295552	Interventional	Evaluation the effect of novel DNA demethylating agents in the treatment of metastatic TNBC (drugs: decitabine, carboplatin).	Partial response (PR) + complete response (CR) rate.	Pathologically confirmed metastatic triple-negative breast cancer, age between 18 years and 70 years.
NCT02957968	Interventional	Course of immunotherapy consisting of sequential decitabine followed by pembrolizumab administered prior to a standard neoadjuvant chemotherapy regimen for patients with locally advanced HER2-negative breast cancer.	Determination of whether the immunotherapy increases the presence and percentage of tumor and/or stromal area of infiltrating lymphocytes prior to initiation of standard neoadjuvant chemotherapy.	Invasive adenocarcinoma of the breast, HER2-negative.
NCT03282825	Interventional	Decitabine and paclitaxel combination therapy in treating patients with metastatic and locally advanced breast cancer.	Measure of maximum tolerant dose (MTD) and dose limiting toxicity (DLT).	Unable to operate for therapy with HER2 negative breast adenocarcinoma and metastatic breast cancer, one or more chemotherapy.
NCT01194908	Interventional	Reactivation of ER using decitabine in combination with LBH589 (deacetylase inhibitor). Reactivated ER can then be targeted with agents such as tamoxifen.	Measure of MTD of Decitabine and LBH589 given in combination and determination of the safety of tamoxifen in combination with decitabine and LBH589.	ER-, PR-, HER2- metastatic or locally advanced breast cancer.
**FdCyd**				
NCT00978250 and NCT01479348	Interventional	Testing of FdCyd (also called 5-fluoro-2′-deoxcytidine), and THU (also called tetrahydrouridine) effectiveness in treating cancer that has not responded to standard therapies.	Determination of PFS and/or the response rate (CR + PR) of FdCyd.	Individuals who were 18 years of age and older who have advanced non-small cell lung cancer, breast cancer, bladder cancer, or head and neck cancer that has progressed after receiving standard treatment.

In the case of breast cancer, DNMTi can be used as an adjunct therapy in the treatment of advanced and hormone-resistant breast cancer. The studies that have been conducted have shown that DNMTi reduce the size of breast tumors and inhibit their growth, in addition to improving the effectiveness of hormonal therapy. In addition, DNMTi may also contribute to the inhibition of the development of metastases in other organs, which may improve the overall effectiveness of treatment. However, DNMTi therapy is still in the clinical trial stage, and its effectiveness in the treatment of breast cancer requires further research and confirmation.

## 5. Changes in Histone Modifications

Covalent modifications of histones most often concern the N-terminal tails, which can undergo various post-translational changes including the following: methylation, acetylation, ubiquitylation, sumoylation and phosphorylation at specific residues [99]. These are key changes in the regulation of cellular processes such as transcription, replication and genome repair. Depending on which residues are modified and the type of modifications present, histone modifications can result in activation or repression. The global consequence of HDAC-mediated loss of acetylated H4-lysine trimethylation is gene repression. In turn, alteration of the methylation pattern, especially of H3 histones, mediated by HMT, has been associated with abnormal gene silencing [100]. The most common histone modification is acetylation, a dynamic event carried out by histone acetyltransferases (HATs) and the deacetylase complex histone factor (HDAC). These enzymes, respectively, add and remove acetyl groups from the tails of [101].

BRCA1 is involved in numerous histone modifications leading to the modulation of chromatin activity. Of these, the most common are: histone H2A ubiquitination (resulting in satellite DNA overexpression, homologous recombination disorders, genomic instability) [102,103,104], deacetylation of H2A and H3, interactions with HDAC1 resulting in deacetylation of DNA repair genes (e.g., *KDM5B*, *Ku70* pathway genes) [105,106,107,108]. In addition, BRCA1 interacts with *CBP* and *p300*, two structurally related HATs [101]. Zheng et al. [109] described the mechanism of estrogen receptor repression through the interaction of the *BRCA1* gene with the catalytic subunits of the deacetylase complex histone. *BRCA1*-dependent ERα repression is largely restored by the HDAC inhibitor trichostatin A. Moreover, *BRCA1* and HDAC2 interactions have an effect on the epigenetic silencing of mir-155 [110]. Silencing of miR-155 expression occurs upon the binding of *BRCA1* to the oncomiR promoter. The consequence of the combination is the recruitment of HDAC2 to deacetylate histones H2A and H3. It has been experimentally shown that the administration of HDAC inhibitors restores the normal level of miR-155 expression. However, this is not possible in BRCA1-deficient cells because a mutation in the BRCT domain prevents interaction with HDAC2 [110]. Loss of histone acetylation may be the cause of gene silencing abnormalities. Therefore, it was considered that treatment with HDAC inhibitors, leading to the restoration of normal histone acetylation patterns, may have anti-cancer effects, following growth arrest, activation of apoptosis and induction of differentiation. Thanks to the introduction of HDAC inhibitors into cancer cells, it is possible to reactivate silenced tumor suppressor genes. Recently, HMT inhibitors (e.g., DZNep) are also being actively investigated.

There are four groups of HDACs: (1) short-chain fatty acids, e.g., sodium butyrate and valproic acid; (2) hydroxamic acids, e.g., trichostatin A, vorinostat, Panobinostat; (3) cyclic tetrapeptides, e.g., depsipetide, romidepsin / isostax; (4) benzamides, e.g., entinostat, tacedynalin [94,111]. Numerous HDAC inhibitors have also been tested to date, but only vorinostat and romidepsin (the treatment of lymphoma skin with T cells) [112,113,114,115,116,117]. Nevertheless, the clinical use of these epidrugs is acceptable in a number of solid tumor cancers, including breast cancer (Table 2), due to in vitro and in vivo antitumor activity [118,119,120,121,122]. HDAC monotherapy has been shown to have a positive effect on apoptosis induction, growth arrest, and differentiation of breast cancer [94,111,118,119,123]. Although numerous histone modifications typical of breast cancer have been described in the scientific literature, the positive effect of monotherapy with inhibitors has not yet been unequivocally proven. Much more promising are the effects of sensitizing cancer cells to radiotherapy and conventional anticancer drugs [94,124]. In overcoming drug resistance, including HER2-targeted therapies, the effectiveness has been confirmed, among others, by for valproic acid, trichostatin A and entinostat [125]. Huang et al. [125] tested the efficacy of SNDX-275, a class I HDAC inhibitor, to overcome trastuzumab resistance in erbB2 overexpressing patients. The mechanism of overcoming drug resistance involved a radical reduction of *erbB3* and its phosphorylation (*P-erbB3*) and inhibition of Akt signaling. The combination of trastuzumab therapy and SNDX-275 significantly enhanced DNA fragmentation, induction of PARP cleavage and caspase-3 activation. The synergy effect was observed in both sensitive and resistant cells [125]. A similar effect was also observed in tamoxifen-resistant cells. Activation of *ESR1* gene expression by HDACi sensitized cells to estrogen receptor-targeted therapy [126,127,128,129]. Unfortunately, the opposite effect is also observed, mainly due to the non-selective effect of HDACi on non-histone proteins. Fisk et al. [130] reported the effects of increased acetylation of heat shock proteins after the administration of verinostat. Hyperacetylation of *hsp90* inhibits its protective function. As a result of this modification, the level of ERalpha decreased in cancer cells and increased ubiquitination. Munster et al. [131] analyzed the effect of vorinostat on tamoxifen therapy. In their study, they assessed histone acetylation and HDAC2 expression in peripheral blood mononuclear cells. From these, they observed significant clinical benefits for patients that correlated with histone hyperacetylation and higher baseline HDAC2 levels. Yardley et al. [132] reported the effect of an HDAC inhibitor on overcoming resistance to hormone therapies in estrogen receptor positive breast cancer. For this purpose, they conducted phase II clinical trials evaluating entinostat in combination with exemestane in patients with metastatic breast cancer. Based on their observations, they concluded that changes in the acetylation pattern may be of clinical benefit in ER-targeted therapies, as the combination of exemestane with entinostat significantly improved progression-free survival and overall survival [132]. Ramaswamy et al. [133] analyzed the efficacy of vorinostat with paclitaxel and bevacizumab in a phase I/II study involving 54 patients with metastatic breast cancer. They observed a significant effect of vorinostat on the improvement of treatment results (49% objective response rate (partial + complete remission) and 78% clinical benefit rate (objective response + disease stabilization > 6 months), with not too severe side effects. As a mechanism of tumor cell sensitization, the authors indicate the induction of histone and alpha tubulin acetylation, and epigenetic inhibition of Hsp90 function [133].

In the treatment of breast cancer, HDACi can be used both as stand-alone drugs and in combination with other anti-cancer therapies. Studies have shown that HDACi have the ability to induce apoptosis of breast cancer cells, which contributes to the reduction in tumor volume. They can also act synergistically with hormonal drugs and increase their effectiveness. HDACi can also be used in the therapy of breast cancer associated with mutations in the *BRCA1* gene, which is involved in DNA damage repair. Research suggests that HDACi may contribute to increasing the sensitivity of cells to therapy, leading to a reduction in tumor size.

Most of the current histone deacetylase inhibitors have limited efficacy against solid tumors and can produce toxic side effects leading to drug resistance. As a result, there is a need for the development of new histone modification inhibitors with improved anti-tumor activities and reduced toxicities for breast cancer therapy, as well as further investigation into their mechanisms of action. In addition to the inhibitors described above, other promising therapeutic agents such as histone methyltransferase (KMT) and histone demethylase (KDM)are being investigated in in vivo and in vitro studies for their potential in treating various types of cancers, including breast cancer. KMT is an enzyme that transfers a methyl group to the lysine or arginine position of histones, which can lead to gene transcription activation or repression [134,135]. One KMT inhibitor that has shown efficacy in mouse breast cancer studies is GSK-J4. Yan et al. [136] demonstrated that GSK-J4 had the ability to effectively inhibit breast cancer stem cells by reducing their expansion, self-renewal capacity, and expression of stemness-related markers. The SETD8 inhibitor has been investigated as a potential therapy for breast cancer associated with the *BRCA1* mutation. New findings suggest that SETD8 and the methylation of its corresponding histone H4K20 play a role in determining the choice of DNA double-strand break repair pathway [137]. KDM, on the other hand, removes methyl groups from histones, which can also affect gene activity [138]. The KDM4 family, including KDM4A-F, plays a role in oncogene activation, tumor suppressor silencing, and chromosomal instability, making them potential therapeutic targets. Inhibitors targeting KDM4 enzymes have shown anticancer effects in vitro; however, their structural similarities and active domain conservation pose challenges in discovering selective inhibitors. Despite promising results, no KDM4 inhibitors have entered clinical trials yet [139]. One KDM inhibitor that shows potential in breast cancer treatment is also GSK-J1 [140]. Wang et al. used GSK-J1, a small-molecule inhibitor of JMJD3/KDM6B to treat LPS-induced mammary gland inflammation in mice and mouse mammary epithelial cells in vivo and in vitro. The KDM1A (LSD1) inhibitor has been investigated as a therapeutic target in breast cancer because it has shown promising results in inhibiting the growth and invasion of cancer cells [141]. However, research on KMT and KDM inhibitors in breast cancer therapy is still in its early stages and requires further investigation to confirm their efficacy and safety. All of the inhibitors mentioned are being currently investigated as single therapies; however, research indicates that combinations of KMT and KDM inhibitors may prove more effective in treating breast cancer (Li et al., 2021) [142]. It is worth noting that unlike traditional chemotherapy, which kills both cancerous and healthy cells, KMT and KDM inhibitors directly affect molecular processes associated with cancer, which can lead to more targeted and effective breast cancer therapy.

## 6. The Role of miRNAs

miRNA expression pattern is associated with oncogenesis. MiRNAs are a special class of small (18–26 nucleotides), evolutionarily conserved non-coding RNA molecules, which, through post-transcriptional regulation of gene expression, affect the proper development and maintenance of tissue homeostasis, as well as being involved in the development of pathological processes [143,144,145]. They are thought to be responsible for regulating close to 30% of human mRNAs; one miRNA can regulate multiple target sequences, and one gene can be controlled by numerous microRNAs. In recent years, they have gained popularity as non-invasive biomarkers for assessing tumor development, as their significant role in carcinogenesis, cancer progression, cell cycle checkpoint bypass, drug response regulation, and invasion has been described [146,147,148,149,150,151] (Figure 2). It has been shown that miRNAs act as both oncogenes and tumor suppressor genes, which gives them a wide range of modulation processes in cancer cells [152,153,154]. Numerous studies have identified miRs associated with abnormal expression and function of breast cancer predisposition genes (Figure 3).

Research on the importance and potential of miRs in oncology is extremely intensive; however, most breast cancer clinical trials are still observational (Table 3). Their goal is to identify miRs that modulate the response of cancer cells to specific drugs, or to analyze the relationship between the selected miRs and the effectiveness of neoadjuvant and adjuvant chemotherapy. Both in scientific and clinical research, it was possible to identify miRs that may be markers of drug resistance or predictors of hormonal sensitivity, as well as molecules associated with the risk of developing complications of therapy.

In this review, a selective choice of presented microRNAs related to breast cancer therapy and monitoring, drug resistance, serving as targets for targeted therapy, and complementing standard breast cancer treatment was made. To maintain an appropriate scope of the review, we focused on microRNAs that can supplement standard breast cancer treatment and have the potential for use in therapeutic protocols. Their significance and prospects for application in clinical practice were presented.

MicroRNAs are currently an important area of research in cancer therapy. The use of miRNAs in breast cancer therapy aims to impact the proliferation, invasion, migration, angiogenesis, and apoptosis of cancer cells. MiRNAs can have the following goals and types of applications in oncology therapy:(1)Diagnosis: miRNAs can be used as biomarkers for assessing the risk of breast cancer development, diagnosis, and disease monitoring [155,156,157,158,159].(2)Targeted therapy: miRNAs that are expressed in cancer cells in large quantities can be inhibited by specific miRNA inhibitors (antimiRNAs). AntimiRNAs are small molecules that bind to the target miRNA and inhibit its activity, leading to the inhibition of the growth and proliferation of cancer cells [160,161,162].(3)Gene therapy: the introduction of miRNAs into cancer cells that express a low level of a particular miRNA can be a method of gene therapy. In gene therapy, miRNAs can be used for therapeutic or diagnostic purposes [163,164,165].(4)Combined therapy: miRNAs can be used in combination with other anticancer drugs to increase their efficacy. For example, miRNAs and chemo- or immunotherapeutic drugs can be used together to increase the effectiveness of therapy and reduce the toxicity of the drugs [166,167,168,169].

In the therapy of breast cancer, various miRNAs play significant roles, including the following:(1)MiR-21: this is one of the most commonly identified miRNAs associated with breast cancer. MiR-21 has been shown to contribute to breast cancer development and progression by regulating a range of genes responsible for proliferation, angiogenesis, and invasion of cancer cells [170,171,172,173,174].(2)MiR-34a: miR-34a has been shown to inhibit proliferation and induce apoptosis of breast cancer cells, making it a potential therapeutic target [175,176,177].(3)MiR-155: it has been shown that miR-155 is involved in inflammatory processes and proliferation of cancer cells in breast cancer. Therefore, miR-155 has become a subject of interest as a potential therapeutic target [178,179,180,181,182].(4)MiR-200: this is a miRNA that is involved in the invasion and metastasis of breast cancer through the regulation of EMT (epithelial-mesenchymal transition) processes. Increasing miR-200 expression has been shown to reduce the ability of cancer cells to invade and migrate, making it a potential therapeutic target [183,184].(5)MiR-10b: it has been shown that miR-10b is involved in the metastasis of breast cancer through the regulation of EMT processes. Therefore, miR-10b has become a subject of interest as a potential therapeutic target [185,186,187].

One of the most extensively studied is the suppressor miR-34a [188]. MiR-34a is one of the most important factors involved in the regulation of cell cycle and apoptosis. Laboratory studies have shown that miR-34a is often mutated or decreased in breast cancer. The use of miR-34a in breast cancer therapy involves introducing an additional dose of miR-34a into cancer cells, which leads to the inhibition of proliferation, invasion, migration, and induction of apoptosis. Studies have shown that miR-34a can also enhance the effect of chemotherapy in breast cancer. For example, in one study, breast cancer cells were treated with a combination of synthetic miR-34a and 5-fluorouracil (5-FU), a chemotherapy drug commonly used in breast cancer treatment. The results showed that the combination of miR-34a and 5-FU led to a greater inhibition of cell growth and increased apoptosis compared to treatment with 5-FU alone. Adams et al. [189] demonstrated the benefits of miR-34a replacement therapy in retarding the growth of subcutaneous and orthotopic transplanted tumors. They have experimentally proven that in triple-negative breast cancer models, restoring miR-34a expression leads to the inhibition of proliferation and invasion as well as the activation of senescence. Moreover, they described a negative feedback between miR-34a and *c-SRC* affecting the sensitization of cancer cells to dasatinib [189]. MiR-34a also has an effect on 5-fluorouracil chemotherapy as shown by Li et al. [190]. In their study, they noticed a significant reduction in the level of miR-34a in breast cancer cell lines and breast cancer samples, which resulted in no inhibition of cancer cell invasion and inhibition of apoptosis. The beneficial effect, including sensitization of cells to 5-FU after administration of miR-34, was due to the effect of targeting *Bcl-2* and *SIRT1*. These studies confirmed the efficacy of the mimetic miR-34a (MXR34) as a potential therapeutic agent for patients with breast cancer [190] and other types of cancer [191,192,193,194]. Li et al. [195] observed that the expression of miR-34a in tissues and drug-resistant cell lines is significantly reduced and correlated with breast cancer multidrug resistance (MDR). As noted by the authors, patients with low miR-34a expression had worse overall survival and disease-free survival; the introduction of the mimetic miR-34a in vitro led to a partial reversal of MDR. The authors indicated *Bcl-2*, *CCND1* and *NOTCH1* as the targets of this miR, while also noting the lack of a direct effect of miR-34a on the expression of *HER-2*, *TP53* and *TOP-2a*. One of the challenges in the use of miR-34a in breast cancer therapy is the problem of delivering miRNA to cancer cells. Currently, various methods of miRNA transport are being investigated, such as lipid nanoparticles, viral vectors, or vector-based nanoparticles, which aim to improve the effectiveness of delivering miRNA to cancer cells.

Xue et al. [169] identified elevated miR-621 expression associated with paclitaxel and carboplatin (PTX/CBP) sensitivity. Administration of the mimetic miR-621 made it possible to sensitize breast tumors to these drugs by inhibiting *FBXO11* and enhancing p53 activity [169]. Mei et al. [196] showed that overexpression of miR-21 is associated with taxol resistance, and Chen and Bourguignon [197] showed that upregulation of this molecule has an effect on the increase in *Bcl-2* activity. Shaban et al. [198] highlighted that miR-34 and miR-21 can predict the response of breast cancer patients to chemo-radiotherapy, especially by regulating *Bcl-2*, *BRCA1*, *BRCA2*, and *p53* targets in breast cancer cells. In turn, Yadav et al. [199] observed a decrease in miR-21 expression in breast cancer patients who received neoadjuvant therapy. Differential expression of miR-125b and changes in miR-21 expression during neoadjuvant chemotherapy were associated with response to chemotherapy and disease-free survival. The correlation of downregulation of miR-125b expression with resistance to four cycles of neoadjuvant 5-fluorouracil, epirubicin and cyclophosphamide was confirmed by Wang et al. [200]. The mechanism of action involved modulation E2F2 expression. MiR-125b is involved in regulating numerous signaling pathways, including *NF-κB*, *p53*, *PI3K/Akt/mTOR*, *ErbB2*, *Wnt*. Hence, it is also important marker involved in controlling cell proliferation, differentiation, metabolism, apoptosis, drug resistance and tumor immunity [201]. In HER2-negative patients, the differential expression of miR-222, miR-20a and miR-451 in clinical response to neoadjuvant chemotherapy has been confirmed to be related to chemosensitivity [202].

Cardiotoxicity is one of the most significant complications among breast cancer patients as a result of the use of agents such as anthracyclines and monoclonal antibodies directed against HER2. Therefore, the search for prognostic biomarkers of this event is crucial in mitigating the risk of cardiotoxicity in vulnerable patients [203,204]. Numerous microRNAs are considered among the candidates, including hsa-miR-1273 g-3p and hsa-miR-4638-3p (regulating *TGF-β* and *CTGF* pathways, responsible for atherosclerotic plaque instability and heart failure [205]), hsa-miR-208 (associated with fibrosis and EMT progression, and specifically targeted to the BMP co-receptor, endoglin [206,207]), hsa-miR-130a (associated with cardiomyopathy [208,209,210]), hsa-miR-29a (involved in hemolysis of blood cells [211,212]) or proangiogenic hsa-miR-17-5p, hsa-miR-19a, hsa-miR-378, hsa-Let-7b [208,209] and hsa-miR-126 [207,211]. On the basis of laboratory tests, the concept of introducing antimiRs as a real cardioprotective agent in patients after chemotherapy is being considered.

**Table 3 ijms-24-07235-t003:** Examples of clinical trials taking into account the importance of microRNAs in the treatment of breast cancer.

NCT Number	Study Type	Description	Outcome Measures	Study Population	Publications
NCT03779022	Observational	miRNA and relevant biomarkers of BC patients undergoing neoadjuvant treatment	Clinical disease response was evaluated for every two cycles of chemotherapy till surgery with RECIST criteria.	Patients with early stage breast cancer patients, with stage II-III disease	[200,202,213,214]
NCT01598285	Observational	Genome-wide association study (GWAS) and microRNA (miRNA) profiling for identification of genetic variants and blood miRNA signatures predictors of bevacizumab response cancer	To identify miRNA signatures in whole blood as bevacizumab response predictors in metastatic breast cancer patients	Patients suffering from metastatic (disseminated at the time of diagnosis) breast cancer, treated with bevacizumab.	[98]
NCT02656589	Observational	A perspective study of the predictive value of microRNA in patients with HER2 positive advanced stage breast cancer who were treated with herceptin	Progression-free survival (PFS) evaluation defined as the interval from the diagnosis of advanced breast cancer with HER2 positive to disease progression, relapse, death due to any causes or last follow-up. The follow-up interval is 2 years.	Advanced breast cancer patients first diagnosis, ≥ 18yrs ages, HER2 positive: immunohistochemistry (+++) or FISH (+), stage IV, the patients have no history of chemotherapy, hormone therapy, radiotherapy or surgery after diagnosis of breast cancer.	[215,216,217]
NCT01612871	Interventional	Circulating miRNAs as biomarkers of hormone sensitivity in breast cancer	Analysis of the larger-scale circulating miRNAs in plasma of these patients before and after one month of treatment with tamoxifen or anti aromatase. Correlation between the specific miRNAs initial expression and the appearance of an objective response or clinical benefit of hormone therapy and the time to progression.	Drugs: Tamoxifen, Letrozole, Anastrozole, Exemestane Women with metastatic invasive breast cancer or locally advanced (without surgical project), for which treatment with tamoxifen or anti aromatase. Cancer HER2-negative.	[98]
NCT05151224	Observational	Circulating microRNA-21 expression level before and after neoadjuvant systemic therapy in breast carcinoma	Describes miRNA 21 expression level before and after neoadjuvant systemic therapy in breast cancer patient.	Invasive breast cancer, from stage IIB to stage IIIC, all subtypes are included, either HR (ER, PR)-positive or -negative, HER2-positive or -negative, eligible to neoadjuvant systemic therapy. Neoadjuvant systemic treatment composed of anthracyclines-based chemotherapy and taxanes, trastuzumab for HER2-positive patients.	[196,197,199,200,202,218,219,220]
NCT01722851	Observational	Novel breast cancer biomarkers and their use for guiding and monitoring response to chemotherapy	Relationship between changes in a patients circulating miRNA expression levels over the course of their systemic therapy, and their response to that treatment.	Cohort 1: All patients with a new diagnosis of breast cancer, who are destined to undergo neoadjuvant chemotherapy. OR Cohort 2: All breast cancer patients who present with metastatic disease, disease recurrence or progression who will receive up-front chemotherapy ± hormonal therapy. OR Cohort 3: All breast cancer patient who present with metastatic disease who are commencing hormonal therapy only.	[98]
NCT02950207	Observational	Prospective observational study of antitumor activity correlation between hormonal therapy and expression miRNA-100	Mono-centric, observational, prospective study, designed for patients with diagnosis of hormone-positive breast cancer to evaluate the correlation between the response to hormonal treatment indicated by the reduction of the level of Ki67 and miRNA100 in two groups of patients	Post-menopausal hormone-positive breast cancer patients. Histological diagnosis of invasive carcinoma of the breast. X-ray evidence (mammography and / or ultrasound) strongly suggestive for the presence of invasive breast cancer (*BIRADS 4c* or *BIRADS 5*) of greater than 15mm diameter. Positivity for the estrogen receptor and / or to the progestin defined as the expression of one or both hormone receptors in ≥10% of tumor cells, negativity for HER2.	[98]

Clinical trials using miRNA in breast cancer therapy are still in the early stages. Most of them have focused mainly on Phase I and II clinical trials aimed at evaluating the safety and tolerability of miRNA. However, the results of these trials are promising, suggesting potential therapeutic benefits and the safety of miRNA use in breast cancer treatment. Several miRNAs, such as miR-34a, miR-21, miR-155, miR-29b, and miR-16, have been identified as potential therapeutic targets in breast cancer therapy. Introducing miRNA as a new therapeutic tool may help to better understand pathological processes and result in improved therapeutic outcomes for patients with breast cancer.

## 7. Summary

The clinical trials conducted to date have failed to confirm the high effectiveness of monotherapy based on epigenetic drugs. According to Falahi et al. [111], anticancer efficacy of epitherapy in breast cancer is at the level of 10%. However, much better results are obtained when the therapy is combined with cytostatics or targeted therapy. Significantly higher progression-free survival and survival were observed in the phase I and II studies.

The introduction of epigenetic drugs for oncotherapy raises questions about their possible side effects. It is believed that epitherapeutics primarily affect rapidly dividing cells; therefore, their toxic effect on normal cells should be minimal. Nevertheless, an alternative approach is actively sought to develop non-nucleoside compounds that can, e.g., effectively inhibit DNA methylation without being incorporated into DNA, such as SGI-1027, RG108 and MG98 [221,222,223]. The mechanism of action of these molecules involves blocking the DNMT catalytic/ cofactor binding sites, or targeting their regulatory messenger RNA sequences. Non-nucleoside DNMTs of natural origin such as polyphenols or epigallocatechin-3-gallate are also tested [87,224,225]. It has been shown that these compounds have, e.g., inhibition effect, VEGF inhibition and apoptosis induction, and *ESR1* re- expression. Nevertheless, the proliferation-inhibiting potential of drugs based on non-nucleoside compounds is not satisfactory. The advantage of miRNAs is that, as natural cellular components, they should not cause side effects and toxicity, and subsequent studies prove high efficiency and low antigenicity of such therapy. However, the clinical use of miRs in breast cancer therapy still requires a significant amount of research work, including refining methods of therapeutic manipulation, delivery to target cells, overcoming immune barriers and maintaining long-term activity of nanoconstructs [66].

Epigenome characteristics can be important in determining prognosis and can be used to stratify patients into risk categories. It is also helpful in identifying breast cancer patients who are likely to respond well to neoadjuvant and adjuvant chemotherapy, or who are sensitive or resistant to a given therapeutic agent. Monitoring individual epimutations or circulating microRNAs could improve patient response to chemotherapy and hormone therapies.

Current breast cancer therapy protocols do not allow the introduction of epigenetic drugs as monotherapy. Such therapy is in its early stages, and requires careful study of its benefits for patients, potential side effects, interactions with other drugs and the exact mechanisms of both the effect on the cancer cell and the acquisition of resistance by it. Much more promising are the possibilities of using epi-drugs in combination with chemotherapeutics and targeted therapies to increase or restore sensitivity to these drugs.

## Figures and Tables

**Figure 1 ijms-24-07235-f001:**
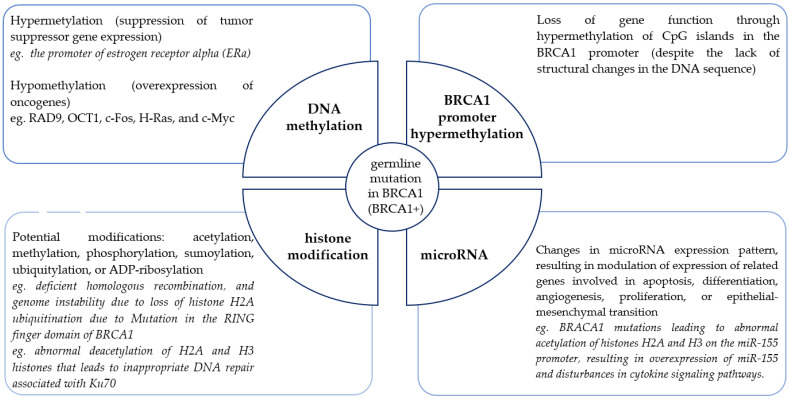
Landscape of epigenetic changes in familial breast cancer associated with the *BRCA1* gene (based on [56]).

**Figure 2 ijms-24-07235-f002:**
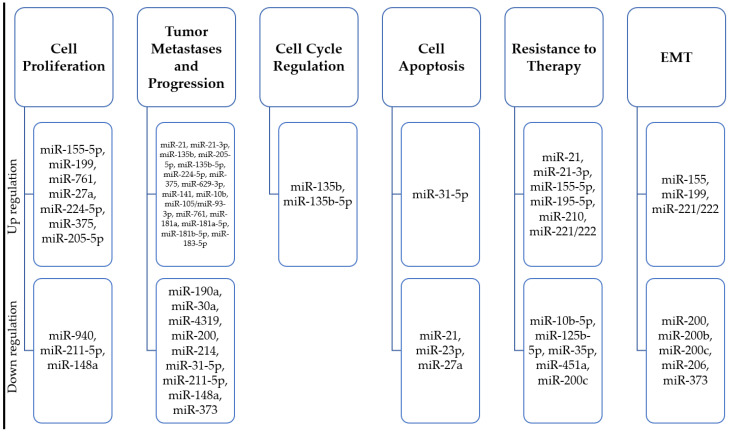
Summary of microRNA molecules characteristic of different stages of progression of hereditary and familial breast cancer.

**Figure 3 ijms-24-07235-f003:**
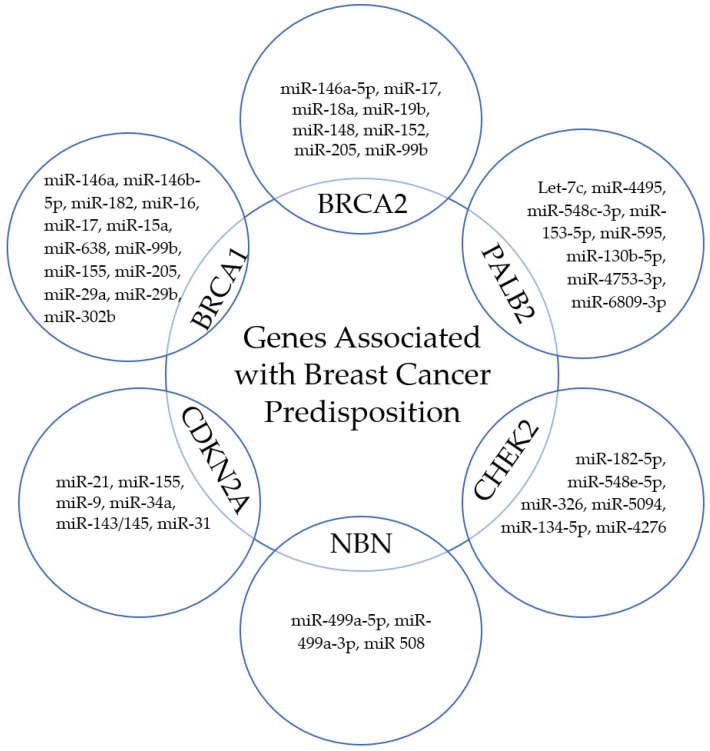
Selected miRs correlated with the expression and function of breast cancer predisposition genes.

**Table 2 ijms-24-07235-t002:** Exemplary clinical studies on the role of histone deacetylase inhibitors in breast cancer (based on ClinicalTrials.gov (accessed on 5 March 2023) [98]).

NCT Number	Study Type	Description	Outcome Measures	Study Population
**Vorinostat**				
NCT00574587	Interventional	Determination of the optimal dose of vorinostat to use in combination with standard chemotherapy alone (or in combination with plus trastuzumab for HER2-positive disease), and to determine whether vorinostat enhances the effectiveness of standard chemotherapy (+/− trastuzumab) in patients with locally advanced breast cancer.	Measure of pathological complete response	Histologically or cytologically confirmed adenocarcinoma of the breast associated with the following stages: IIB, IIIA, IIIB or IIIC; Her2/neu positive; no prior chemotherapy, radiation or definitive therapeutic surgery
NCT01084057	Interventional	Determination of the safety and tolerability of the combination of vorinostat with ixabepilone	Objective response rate and/or clinical benefit rate; toxicity profile	Histologically or cytologically confirmed stage IV adenocarcinoma of the breast
NCT03742245	Interventional	Testing of the safety and preliminary efficacy of 10laparib and vorinostat when used together in participants with relapsed/refractory and or metastatic breast cancer.	MTD, dose-limiting toxicities, RP2D and antitumor activity	Breast cancer with the exception of human epidermal growth factor receptor 2-positive breast cancer.
NCT01194427	Interventional	Looking at the effects of the combination of vorinostat (Suberoylanilide Hydroxamic Acid or Zolinza) and tamoxifen on breast cancer tissue.	Determination of the percentage change in proliferation index Ki-67 in both ER+ and ER- tumors between baseline and post-treatment biopsy	Stage I-III invasive breast cancer
NCT01153672 Additionally, NCT01720602	Interventional	Treating patients with stage IV breast cancer receiving aromatase inhibitor (AI) therapy.	Determination of the rate of clinical benefit (objective response plus stable disease); duration of response, PFS, overall survival	Histologically or cytologically proven diagnosis of breast cancer.
NCT01695057	Interventional	Evaluation of the ability of HDAC inhibition using suberoylanilide hydroxamic acid (SAHA, vorinostat) to induce expression of the ER and PR genes in solid human triple-negative invasive breast cancer.	Combined PR/ER response, grade 3 or 4 toxicities	Resectable tumor measuring 2 cm or more
NCT00616967	Interventional	Studying how well giving carboplatin together with paclitaxel albumin-stabilized nanoparticle formulation works with or without vorinostat in treating women with breast cancer that can be removed in surgery.	pCR rate, cCR, absolute change from baseline in Ki-67, changes in methylation index within a panel of 10 genes which included: *HIST1H3C*, *AKR1B1*, *GPX7*, *HOXB4*, *TMEFF2*, *RASGRF2*, *COL6A2*, *ARHGEF7*, *TM6SF1*, and *RASSF1A*.	Histologically confirmed infiltrating ductal breast cancer by core needle biopsy, HER2-negative disease
NCT04190056 and NCT02395627	Interventional	Studies how well pembrolizumab (monoclonal antibody) and tamoxifen with or without vorinostat work for the treatment of estrogen receptor positive breast cancer.	Overall response rate, duration of response, PFS and OS	Pre- and postmenopausal women or men with stage IV ER+ breast cancer histological or cytological confirmation
NCT00365599	Interventional	Exploration the efficacy of vorinostat and tamoxifen combined.	OR, time to progression, safety evaluation	Cytologically/histologically documented locally advanced or metastatic breast cancer, ER+ or PR+
**Entinostat**				
NCT04296942	Interventional	Analysis of new combination of immunotherapy drugs in metastatic breast cancer (drugs: entinostat, biological: brachyury-TRICOM, M7824, ado-trastuzumab emtansine).	Overall response, PFS, TILs	Adults 18 and older who have been diagnosed with metastatic breast cancer, such as Triple-negative Breast Cancer (TNBC) or estrogen receptors (ER)-/progesterone receptors (PR)-/human epidermal growth factor receptor 2 (HER2)+ Breast Cancer (HER2+BC)
NCT02115282	Interventional	Evaluation of whether the addition of entinostat to endocrine therapy (exemestane) improves PFS and/or OS in patients with HR+, HER2-negative locally advanced or metastatic breast cancer who have previously progressed on a non-steroidal aromatase inhibitor.	Objective response rate, PFS, OS, time-to-treatment deterioration, lysine acetylation change in CD45 blood mononuclear cells, health-related quality of life	H.istologically confirmed adenocarcinoma of the breast with staining of ≥ 1% cells is considered positive
NCT03473639	Interventional	Determination of` the safety and side effects of combining entinostat, with capecitabine, in both participants with metastatic breast cancer and then participants with high-risk breast cancer after neo-adjuvant therapy.	Identification of a maximum tolerated dose combination of entinostat and capecitabine; frequency of adverse events, DFS, OS, relationship of circulating tumor DNA and residual disease	Histologically confirmed diagnosis of stage IV invasive breast cancer, positive OR negative estrogen and progesterone receptor status.
NCT00676663 and NCT02820961	Interventional	Evaluation of the safety and efficacy of entinostat in combination with exemestane in the treatment of advanced breast cancer.	PFS, ORR, clinical benefit rate	Postmenopausal female patients, ER+, relapsed or progressed on prior treatment with aromatase inhibitor
NCT02453620	Interventional	Evaluate the safety and tolerability of the combination of entinostat and nivolumab with or without ipilimumab in subjects with advanced solid tumors.	Incidence of adverse events, changes in ratio of effector T cell (Teff) to regulatory T cell (Treg) in tumor biopsies, CR, PR, SD, PFS, post-combination therapy expression of checkpoint inhibitors (PD-1/PD-L1) in tumor biopsies, changes in other immune-related biomarkers, analysis of tumor-specific mutations and mutant neo-antigens, Changes in candidate gene re-expression in malignant tissue, gene methylation silencing in circulating DNA and malignant tissue pre and post-therapy, pharmacodynamic outcomes	Confirmed invasive adenocarcinoma of the breast HER2- that is locally advanced/metastatic and has progressed despite standard therapy
**Panobinostat**				
NCT01105312	Interventional	Studying the side effects and best dose of panobinostat when given together with letrozole and to see how well it works in treating patients with metastatic breast cancer.	Maximum-tolerated dose, response rate, survival time, time-to-disease progression, PFS, CR, PR, SD	Any ER, PR, or HER2 level
NCT00788931 and NCT00567879	Interventional	Identification the maximum tolerated dose of both intravenous and oral panobinostat when given in combination with trastuzumab and paclitaxel.	Determination of MTD, safety and tolerability, evaluation of the efficacy	Adult female patients with HER2+ metastatic breast cancer
NCT00777335 and NCT00777049	Interventional	Analysis of the benefit of panobinostat monotherapy given either orally or i.v. to women with HER2-positive locally recurrent or metastatic breast cancer	The assessment of OR, CR + PR	Women with v-ERB-B2 avian erythroblastic leukemia viral oncogene homolog 2 (HER2) positive locally recurrent or metastatic breast cancer
**VPA**				
NCT00395655	Interventional	Analysis of the benefit of the demethylating hydralazine plus the HDAC inhibitor magnesium valproate addition to neoadjuvant doxorubicin and cyclophosphamide in locally advanced breast cancer to assess their safety and biological efficacy.	Global DNA methylation, histone deacetylase activity and global gene expression	Aged 18 and older; histologically proven invasive T2-3, N0-2, and M0 (stages IIB-IIIA) breast carcinoma.
**Depsipeptide/Romidepsin**				
NCT01938833	Interventional	Studies the side effects and best dose of romidepsin when given together with paclitaxel albumin-stabilized nanoparticle formulation and to see how well they work in treating patients with metastatic inflammatory breast cancer.	MTD, PFS, ORR, CBR and incidence of adverse events	Breast carcinoma with a clinical diagnosis of IBC based on the presence of inflammatory changes in the involved breast, such as diffuse erythema and edema
NCT02393794	Interventional	Studies the combination use of cisplatin plus romidepsin and nivolumab in metastatic triple-negative breast cancer or *BRCA* mutation-associated locally recurrent or metastatic breast cancer	MTD and ORR determination, median progression-free survival and overall survival	Confirmed germline *BRCA1* or *BRCA2* mutation, regardless of subtype of breast cancer
NCT00098397	Interventional	Determination of the efficacy and safety of FR901228 (depsipeptide) in patients with metastatic breast cancer.	Clinical activity of this drug, in terms of progression-free survival, in these patients	Metastatic disease, patients have received prior anthracycline (doxorubicin or epirubicin) and/or taxane (paclitaxel or docetaxel) as adjuvant therapy or for advanced disease

## Data Availability

Not applicable.

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
