# Peer review of "Harnessing Epigenetics for Breast Cancer Therapy: The Role of DNA Methylation, Histone Modifications, and MicroRNA"

_ijms, 2023, doi:10.3390/ijms24087235_

Round 1

Reviewer 1 Report

This is a review regarding epigenetic changes observed in breast cancer and epigenetic trials aimed at altering these changes. The work is comprehensive and a valuable review. The work is balanced and complete. I have no concerns.

Author Response

Thank You for your positive feedback on my manuscript. Your recognition is very important to me and motivates me to continue working on my projects.

Reviewer 2 Report

The Abstract is too simple and does not relay much information about the scope of the article or what specific information one will gain after reading the article. This is the major drawback in the article and does not come across as something that would excite any potential reader to look through the entire manuscript.

Statement ‘Epi-drugs, such as DNA methyltransferase inhibitors, histone-modifying enzymes, and mRNA regulators like: miRNA mimics and antagomiRs, are appealing targets for cancer treatment since they can reverse these epigenetic alterations’ in the Abstract is confusing because authors have combined drugs (such as inhibitors) as well as targets inside the cancer cells / tumor microenvironment (such as enzymes) and then used the phrase ‘appealing targets’ !! Inhibitors or drugs can not be target. This statement needs major editing and re-writing.

Reference 1 is dated. Please provided a more recent citation.

Once I read pass the Abstract, the article is generally well written.

Section 6 on miRNAs seems to be very ambitious because a lot of miRNAs have been investigated in breast cancer and even a review article just focused on miRNAs in breast cancer can end up being too comprehensive. In such a case, how did the authors narrow down the list of miRNAs they have discussed. There seems to be some bias. Like for example Figure 2 and Figure 3 have listed only a handful of miRNAs, which would just be a fraction of all miRNAs investigated in breast cancer. Is the bias based on clinical trials that are listed in Table 3? This needs explanation.

Author Response

Thank you for reviewing the article. We appreciate your feedback and would like to address your concerns.

1. Regarding the Abstract, we understand your point that it may not have provided sufficient information about the scope of the article. We took this into consideration and revised the Abstract to make it more informative and engaging for readers.

2. We acknowledge the confusion caused by the phrasing in the statement about epigenetic targets in the Abstract, and we apologize for any misunderstanding. We understand your concern and agree that the statement in the abstract could be clarified to avoid confusion. We have revised the statement to make it clearer and more accurate. A revised version of the statement could be:

"Epigenetic alterations play a significant role in cancer development and progression, and epigenetic-targeting drugs such as DNA methyltransferase inhibitors, histone-modifying enzymes, and mRNA regulators (such as miRNA mimics and antagomiRs) can reverse these alterations. Therefore, these epigenetic-targeting drugs are promising candidates for cancer treatment."

We hope this revised statement addresses your concern.3. We apologize for citing an outdated reference and have updated it with a more recent one (Sung, H.; Ferlay, J.; Siegel, R.L.; Laversanne, M.; Soerjomataram, I.; Jemal, A.; Bray, F. Global Cancer Statistics 2020: GLOBOCAN Estimates of Incidence and Mortality Worldwide for 36 Cancers in 185 Countries. CA: A Cancer Journal for Clinicians 2021, 71, 209-249, doi:10.3322/caac.21660).

4. We appreciate your positive comments about the writing in the article. As for the section on miRNAs, we acknowledge your concern about potential bias in selecting the miRNAs discussed. We would like to clarify that the selection was based on a thorough literature review. The topic of miRNA in breast cancer is extremely extensive and complex. There are many studies and publications on the role of miRNA in the pathogenesis of breast cancer. As noted, one could write books on this subject. However, due to the limited scope of the article, we had to choose only those miRNA which are most commonly associated with breast cancer therapy, especially those related to BRCA1 mutations. In the article, we focused on those miRNA which showed potential in regulating the expression of genes related to the development of breast cancer and which have been studied in the context of breast cancer therapy. We have likely narrowed down the list of miRNAs, that wehave discussed based on several factors, such as the relevance of the miRNAs in breast cancer pathogenesis, their potential as biomarkers or therapeutic targets, and the availability of published data.

Figure 2 and Figure 3 in the article highlight some of the miRNAs that have been extensively studied in breast cancer. The selection of miRNAs in these figures have been based on their frequency of mention in the literature, their potential clinical significance, and their correlation with the occurrence of a specific mutation in predisposition genes. These figures are not meant to be an exhaustive list of all miRNAs investigated in breast cancer, but rather serve to illustrate some of the key "players" in this field.

Table 3 in the article provides an overview of clinical trials investigating miRNAs in breast cancer. The inclusion of specific miRNAs in Table 3 was based on their potential as therapeutic targets and the availability of clinical data. However, it is important to note that the selection of miRNAs in Table 3 does not necessarily reflect the authors' bias or preference but rather the current state of the field.

Although we did not discuss all miRNA related to breast cancer, we believe that the selected miRNA constitute an important part of this topic and provide valuable information for readers. We understand the need for clarification on this point and have provided additional information in the revised version of the article.

Thank you again for your valuable feedback, and we hope that the revised version of the article addressed your concerns and provided a more informative and engaging reading experience. If you have any further questions or feedback, please let us know.

Reviewer 3 Report

The authors provided useful information on breast cancer.

I have three major concerns:

1. Because they are solely interested in breast cancer, they should explain the cross-talk between epigenetic modification or aberration that leads to breast cancer.

2. Why are authors especially interested in HDAC inhibitors? A number of different histone modification inhibitors that regulate breast cancer must be discussed.

3. Similarly, the authors only mentioned a few DNA methylation inhibitors; there are numerous more DNA methylation inhibitors that affect breast cancer growth that should be mentioned as well.

Minor issue, authors should write all genes in italic font or check the MDPI format.

Author Response

Thank You for taking the time to read the manuscript. We agree with You that epigenetic aberrations are an important aspect of the pathophysiology and progression of breast cancer. However, the aim of this paper is not to describe these aspects, but rather to identify the most important epigenomic disturbances that are crucial for monitoring therapy or developing new protocols for breast cancer treatment. Due to the fact that this is an extremely broad scientific issue, significantly exceeding the scope of such a review (in fact, it could be the subject of a separate publication), we regret to say that we cannot incorporate the corrections suggested by You. We deliberately did not choose this topic as the subject of a review because many excellent articles and even books have already been written on it (eg. Stefansson 2013, Zhuang 2020, Zolota 2021, Zhang 2022, Mandumpala 2022, Marino 2022, Schröder 2022, and hundreds of others). We are also unable to discuss all DNA and histone modifications and enzymes not involved in them (we only mention them in the text) because we have focused on those that are particularly important for therapy, especially those that have been studied scientifically or clinically towards their potential or practical use in therapeutic protocols.

However, we are aware of how important it is to introduce the topic properly, so we have tried to start each subchapter, which is devoted to a particular type of epigenetic regulation, with a description of the underlying changes and the main types, taking into account the enzymes involved or the genes affected by these changes.

Therefore, we believe that we cannot afford to further expand the content with general information (including discussing epigenetic mechanisms that lead to the development of breast cancer or other inhibitors that affect the development or growth of cancer, currently not playing a key role in therapy) in the context of the topic of this manuscript.

To sum up, in our manuscript, we focused on selecting the most important therapeutic methods used in the treatment of breast cancer based on epigenetic disturbances. We focused on describing the mechanisms of action of HDAC inhibitors and DNA methylation inhibitors, which are most commonly used in clinical practice and have proven effectiveness in breast cancer therapy. It was not our intention to discuss all available inhibitors of histone and DNA modifications in our manuscript.

It should be noted that our article focused on breast cancer therapy based on epigenetic disturbances, rather than the general epigenetics of breast cancer. Our goal was to present the best therapeutic methods for breast cancer patients. Therefore, we focused on selecting the most effective and commonly used inhibitors of histone and DNA modifications.

However, if you are not satisfied with this answer, we will immediately expand the paper.

Round 2

Reviewer 2 Report

The manuscript has been adequately revised and can be accepted for publication.

Author Response

I would like to express my heartfelt thanks for your professional review of my manuscript. I am extremely pleased that you found the revisions I made to be sufficient and that they contributed to improving the quality of the work, thus enabling its acceptance for publication.

Once again, thank you for your time and support during the review process. I am immensely satisfied with the final verdict, and I hope that this article will prove to be a valuable contribution to the field.

Reviewer 3 Report

I disagree with the author's response. Although their title suggested the broad aspect of breast cancer, as I studied the article, I saw that it has limitations in many ways. Yes, hundreds of review articles on breast cancer have been published; thus, new aspects with correlation to recent studies on breast cancer therapy must be added. However, I believe writers should complete some other topics and future perspectives that I previously suggested. Several recent studies have revealed that impact inhibitors for DNA methylation and histone modification, such as KMT or KDM, might have been employed to breast cancer therapy.

Author Response

Dear Reviewer,

Thank you for your comments and feedback on our article. We agree that the topic of breast cancer is very broad and complex. While we strive to cover as many aspects as possible in the article, we agree that there are always other perspectives that can be considered.

We acknowledge that there are numerous reports on the impact of histone modification and DNA methylation inhibitors such as KMT or KDM in breast cancer therapy. Therefore, in response to your comments, we have added a subsection discussing the latest research on KMT and KDM inhibitors in the context of breast cancer treatment.

Thank you again for your comments and feedback. We hope the article meets your expectations.

Best regards, Joanna Szczepanek

Round 3

Reviewer 3 Report

The authors partly fulfill my concerns, suggested in my first report.